# Pyrolysis Combustion Characteristics of Epoxy Asphalt Based on TG-MS and Cone Calorimeter Test

**DOI:** 10.3390/ma15144973

**Published:** 2022-07-17

**Authors:** Xiaolong Li, Junan Shen, Tianqing Ling, Qingbin Mei

**Affiliations:** 1School of Civil Engineering, Chongqing Jiaotong University, Chongqing 400074, China; xiaolongli1989@foxmail.com (X.L.); lingtq@163.com (T.L.); 2Yunnan Research Institute of Highway Science and Technology, Kunming 650051, China; meiqb@126.com; 3Civil Engineering and Construction Management, Georgia Southern University, Statesboro, GA 30458, USA

**Keywords:** epoxy asphalt, TG-MS, cone calorimeter test, pyrolysis combustion characteristics

## Abstract

To examine the pyrolysis and combustion characteristics of epoxy asphalt, the heat and smoke release characteristics were analyzed via TG-MS and cone calorimeter tests, and the surface morphology of residual carbon after pyrolysis and combustion was observed via scanning electron microscopy. The results showed that the smoke produce rate of epoxy asphalt was high in the early stage, and then sharply decreased. Moreover, the total smoke produced was close to that of base asphalt, and the surface of residual carbon presented an irregular network structure, which was rough and loose, and had few holes, however most of them existed in the form of embedded nonpenetration. The heat and smoke release characteristics of epoxy asphalt showed that it is not a simple fusion of base asphalt and epoxy resin. Instead, they promote, interact with, and affect each other, and the influence of epoxy resin was greater than that of base asphalt.

## 1. Introduction

At present, asphalt pavement is widely used in highway construction in China, and various asphalt modification technologies have been developed. Among them, epoxy asphalt is used for constructing steel bridge surfaces and tunnel pavement because of its high strength, high temperature resistance, and fatigue resistance [1,2,3]. Epoxy asphalt is synthesized from asphalt and epoxy resin. Asphalt is a complex mixture of hydrocarbon and nonhydrocarbon derivatives, and is mainly composed of carbon, hydrogen, and other elements. Epoxy resin (EP) is a polymer material with two or more epoxy groups on the molecular chain and is a widely used thermosetting plastic [4]. Both asphalt and EP are combustible. In case of a fire caused due to fuel leakage, a large amount of heat and toxic smokes are released [5], which severely affect the structural safety of bridges and tunnels [6,7,8], endanger personal safety, and pollute the environment [9,10,11]. Therefore, examining the pyrolysis and combustion characteristics of epoxy asphalt is important for road fire safety and methods of flame retardant and smoke suppression.

Wu et al. [12] used the Rosemount NGA2000 analyzer to analyze the combustion characteristics of asphalt at a high-temperature rise rate and examined the release law of gaseous products of asphalt and its slurry combustion. Zhu et al. [13] studied the combustion mechanism and gaseous products of asphalt binder at five different oxygen concentrations (21%, 18%, 15%, 12%, and 10%) by thermogravimetry-Fourier transform infrared spectroscopy (TG-FTIR), and results show that the influence on the combustion of heavy components is more significant. Shi et al. [14] analyzed the combustion characteristics of asphalt via a thermogravimetry–infrared combined experiment and discussed the dynamic evolution law of asphalt along with a four-component combustion process. Xia et al. [15] studied the composition of gaseous products of four-component combustion of asphalt via thermogravimetry and mass spectrometry (TG-MS). Xia et al. [16] analyzed the thermogravimetric curves of asphalt pyrolysis under different heating rates and found that the heating rate can affect the pyrolysis mass loss, morphology, and total number of products. Yang et al. [17] evaluated the composition and distribution of pyrolysis products of four components in the pyrolysis process by pyrolysis gas chromatograph coupled with mass spectrometry (PY-GC–MS). Huang et al. [18] reported that the mixing speed, heating temperature, and heating duration of asphalt are the main factors that change the asphalt smoke emission. Wang et al. [19] summarized the test methods of fire effluents produced by a bitumen and asphalt mixture after combustion and determined the influencing factors of fume concentration and composition. Mouritz [20] summarized the key problems of thermosetting matrix composites’ fire behavior of epoxy and phenolic resins, including combustion mechanism, flame retardant, and fire reaction characteristics. Wen et al. [21] studied the pyrolysis characteristics of epoxy resin in SF_6_/N_2_ environment via molecular dynamics. Li et al. [22] studied the heat release rate (HRR), total heat release (THR), smoke production rate (SPR), and total smoke release (TSR) of epoxy resin based on the cone calorimeter method. Xu et al. [23,24] studied the combustion pyrolysis characteristics of carbon fiber/epoxy composites via thermogravimetric analysis and the cone calorimeter method. Zhong et al. [25] used Amsterdam density functional (ADF) software to simulate the correlation between the formation of oxygen-containing small molecules and the number of hydroxyl radicals at different temperatures during resin pyrolysis. It can be seen that researchers all around the world have explored the pyrolysis and combustion properties of asphalt and epoxy resin; however, these were all independent analyses of materials, and there was little discussion on the synthesized epoxy asphalt. Therefore, further research is needed.

According to the composition characteristics of epoxy asphalt, the heat and smoke release characteristics of epoxy asphalt are analyzed via TG-MS and cone calorimeter tests, and the surface morphology of residual carbon after pyrolysis and combustion can be observed via scanning electron microscopy (SEM), which provides a theoretical basis for further analysis of the pyrolysis and combustion characteristics of epoxy asphalt.

## 2. Materials and Methods

### 2.1. Raw Materials

The base asphalt adopts Shell Pen 70, and the main performance indexes are listed in Table 1. The test results meet the specification requirements [26]. Epoxy asphalt was prepared by premixing component A (E-51 bisphenol A type) and component B (modified aromatic amine curing agent) in a ratio of 60:40 (mass ratio) at 60 °C to first form the epoxy resin and was then poured into equal mass base asphalt at 150 °C for 4min of shear mixing. The main performance indexes of component A and epoxy asphalt are listed in Table 2 and Table 3, respectively. The test methods, conditions, and technical requirements for performance indexes of epoxy resin and epoxy asphalt shall be implemented in accordance with the specification [27]. The samples of epoxy resin and epoxy asphalt were first cured in an oven at 150 °C for 3 h, followed by curing in an oven at 60 °C for 4 days. It was finally placed at 25 °C for 1 day before testing.

### 2.2. Test Methods

(1)TG-MS

In this study, the pyrolysis characteristics of base asphalt, epoxy resin, and epoxy asphalt were examined using the Japanese physio-thermo plus EV2/thermo mass photo TG-MS technology under a 70 eV electron ionization (EI) source. The heating range of TG ranged from 25 to 800 °C and the heating rate was 10 °C/min. The thermogravimetric analyzer and mass spectrometer were used simultaneously, and the scanning mode was ion scanning, simulating air atmosphere.

(2)Cone Calorimeter Test

The cone calorimeter test is one of the most commonly used test instruments to study the combustion performance of materials. It is capable of examining time to ignition (TTI), HRR, THR, SPR, TSR, and gas parameters under different oxygen mole fractions, and is widely used in the evaluation of the combustion performance of materials. In this test, the FTT0402 cone calorimeter manufactured by the Fire Testing Technology Company in the UK was used for the mesoscale combustion test. The tests conducted were in accordance with ISO 5660-1 and ASTM E1354 standards. Each sample was wrapped with aluminum foil paper and was tested under 50 kW/m^2^ of thermal radiation. The sample specification was 100 × 100 × 3 mm^3^.

## 3. Results and Analysis

### 3.1. TG-MS Analysis

#### 3.1.1. TG Analysis

TG and differential thermal gravity (DTG) of epoxy asphalt are shown in Figure 1.

It can be seen from the figure that the weight loss of epoxy asphalt mainly occurs at 232.1–579.9 °C. Moreover, the weight loss rate is 96.13% and the maximum weight loss rate is −6.093 %/min. According to the weight loss rate curve, the pyrolysis weight loss process of epoxy asphalt can be divided into three consecutive stages. The temperature in the first stage ranges from 232.1 to 387.9 °C and the corresponding T_max1_ is observed at 342.8 °C, and the weight loss is approximately 38.1%. In this stage, the weight loss mainly occurs due to the fracture and degradation of the network structure of epoxy resin, the large amount of pyrolysis weight loss of saturated components, and participation of some small molecules of volatile aromatic components during pyrolysis. The saturated fraction and aromatic fraction are decomposed into alkanes and cycloalkanes under high-temperature conditions. With the further increase in temperature, the alkyl side chain breaks to generate some short-chain alkanes and gaseous products [28]. The temperature in the second stage ranges from 387.9 to 467.3 °C and the corresponding T_max2_ is observed at 440.8 °C, and the weight loss is approximately 23.8%. This stage mainly involves the decomposition of compounds and colloids formed in the previous stage. At higher temperatures, some relatively stable chemical structures begin to undergo cracking reaction and generate a large number of free radicals and a large number of gaseous products. Moreover, dehydrogenation polymerization of residual substances generates more stable macromolecular substances. The temperature in the third stage ranges from 467.3 to 579.9 °C and the corresponding T_max3_ is observed at 510.9 °C, and the weight loss is approximately 33.7%. This stage is mainly caused by the further degradation of semicoke polymer formed in the first stage of epoxy resin and the pyrolysis of heavy components in asphalt. In this stage, the carbonized layer generated by alkyl side-chain dehydrogenation polymerization in the first and second stages begins to crack and burn, and the heavy substance, asphaltene, in asphalt begins to burn [29].

The pyrolysis process of epoxy asphalt shows obvious multi-stage characteristics, indicating that the pyrolysis reaction process of epoxy asphalt is very complex. This is mainly because the base asphalt and epoxy resin are complex mixtures, and each component has different thermal stability. During the heating process, they gradually start pyrolysis, combustion, absorb, or release different amounts of heat, resulting in the multi-stage characteristics of the DTG curve.

#### 3.1.2. MS Analysis

Figure 2 shows the MS-DTG curve of pyrolysis volatiles of epoxy asphalt, in which the DTG curve provides a reference for analyzing the precipitation process of volatiles. Table 4 lists the distribution of pyrolysis volatiles of epoxy asphalt at each stage. Figure 3, Figure 4 and Figure 5 show the mass spectrum of volatile matter from pyrolysis combustion of epoxy asphalt at characteristic temperatures.

According to the test results of the MS analysis, in the first stage of pyrolysis, the percentage of weight loss was the largest; however, the amount of volatile released was insufficient and mainly included carbon monoxide, water, carbon dioxide, hydrogen peroxide, acetaldehyde, and propane. This is because in the first stage, the fracture and degradation of the epoxy resin network structure occurred with large weight loss; however, the composition of epoxy resin was relatively uncomplicated and less volatile. The third stage comprised the most volatile components, because with the gradual increase in temperature, the analysis of the light component in the saturated component showed that the alkyl side chain broke, there were more high molecular weight components, and the generated coke began to burn and release more volatiles.

It can be seen from Figure 2 that the peak value of the ion flow curve appears when it is close to the peak temperature of each pyrolysis stage, indicating that the volatile release is the largest at this time. Figure 3, Figure 4 and Figure 5 show that the high-content ions in the gas volatiles at the peak temperature in the three stages of pyrolysis are the same, that is, H_2_O with a mass charge ratio of 18, N_2_, CO, and C_2_H_4_ with a mass charge ratio of 28, and CO_2_, C_2_H_4_O, N_2_O, and C_3_H_8_ with a mass charge ratio of 44. Compared with the second stage, the overall ion current intensity in the third stage is larger, and C_3_H_8_O, C_2_H_4_O_2_, C_2_H_8_N_2_, and C_2_H_4_S with a mass charge ratio of 60 and SO_2_ with a mass charge ratio of 64 are also produced.

### 3.2. Cone Calorimeter Test Analysis

The FTT0402 cone calorimeter was used for the mesoscale combustion test. The test results of different materials are listed in Table 5, and the residue after the test is shown in Figure 6.

Under strong thermal radiation, all types of materials quickly burned. The HRR reached the peak rapidly after ignition, and then abruptly dropped. The insufficient combustion of the polymer generated a large amount of black smoke and had a pungent smell. Within 500 s, all materials burned. Among them, the residual carbon remaining after pyrolysis and combustion of epoxy resin burned the least, and the residual carbon rate was less than 1%, which is consistent with the test results of thermogravimetric analysis. It was speculated that most substances in the materials were decomposed into gas components. Although base asphalt and epoxy asphalt had some residual carbon after combustion, the residual carbon was relatively broken and did not form a complete carbon layer.

#### 3.2.1. Combustion Heat Release Performance of Different Materials

The HRR and THR of different materials are shown in Figure 7 and Figure 8, respectively.

TTI determined the flammability of polymer under actual fire conditions. The ignition time of each material ranged between 31 and 36 s. Moreover, resin materials were easier to ignite, and it took 6–8 min for each material to burn completely until it self-extinguished.

HRR and THR are important for estimating the combustion safety of polymers. Both these parameters represent the thermal feedback generated in the material combustion process. The larger the HRR and THR, the more intense the pyrolysis process; however, it also results in worsening the safety. According to the test results, the time for base asphalt, epoxy resin, and epoxy asphalt to reach the peak of HRR was 100, 70, and 60 s, respectively. It was easier for the resin materials to reach the combustion conditions, and the peak heat release rate (pkHRR) and THR of epoxy resin were the highest, reaching 1015.7 kW/m^2^ and 104.5 MJ/m^2^, respectively. The HRR curve presented sharp peaks. Although the pkHRR and THR of base asphalt were the lowest, that is, 328.8 kW/m^2^ and 50.5 MJ/m^2^, respectively, the combustion curve was relatively smooth, indicating that the epoxy resin combustion was more intense, and the base asphalt combustion was less intense.

#### 3.2.2. Combustion Smoke Release Performance of Different Materials

SPR, TSR, CO production rate (COP), and CO_2_ production rate (CO_2_P) of different materials are shown in Figure 9, Figure 10, Figure 11 and Figure 12.

Overall, the trend of SPR was consistent with that of HRR. When the HRR reached the peak, the SPR was the highest. The SPR and TSR of epoxy resin were higher during the test, while the SPR of epoxy asphalt was higher in the early stage, and then sharply decreased. The TSR was close to that of base asphalt, which was approximately 2400 m^2^/m^2^, indicating that the introduction of base asphalt into epoxy resin in the form of an interpenetrating network reduced the smoke generation of resin to a certain extent.

The values of COP and CO_2_P of base asphalt were the lowest, that is, 0.00635 and 0.1639 g/s, respectively. In contrast, the values of COP and CO_2_P of resin materials were higher, which was mainly caused by high-temperature fracture and combustion of the polymer chain.

According to the results of heat release and smoke release tests, the combustion characteristics of epoxy asphalt were not between those of base asphalt and epoxy resin; instead, they promoted, interacted with, and influenced each other. In addition, although epoxy asphalt was the equal mass synthesis of base asphalt and epoxy resin, the pyrolysis and combustion characteristics of epoxy asphalt were closer to that of epoxy resin, indicating that the influence of epoxy resin was greater than that of base asphalt.

### 3.3. SEM Analysis

To further understand the combustion mechanism of different materials, the residual carbon was examined using the cone calorimeter method and was sprayed with gold, and SEM (manufactured by ZEISS Sigma 300, Jena, Germany) was used. The accelerating voltage was 20 kV, amplified 50, 200, and 2000 times to observe the surface morphology. The morphology of residual carbon is shown in Figure 13, Figure 14 and Figure 15.

At high temperature, the surface of base asphalt will flow and agglomerate, resulting in molten droplets, which accelerates the endothermic decomposition of asphalt. After combustion, the surface of residual carbon presented a uniform high-low fluctuation shape, which is a skeleton pore structure and has many through holes with a large pore diameter and loose structure. This is mainly caused by the precipitation of internal decomposition gas breaking through the barrier layer during asphalt combustion. Notably, more through holes indicate more volatile matter in gas phase. Under a high-power microscope (2000× magnification), it can be observed that the residue had a fine-scale structure, which may be formed by stacking collapsed structures after decomposition.

The carbon residue rate of epoxy resin after combustion was low, but the carbon residue surface was dense and flat with few through holes. The main reason for this is that the volatile content was relatively small, and the Figure 14c 2000× magnification figure shows that the combustion form is deepening combustion layer-by-layer at the hole-forming place.

It can be seen from Figure 15 that after the combustion of epoxy asphalt, the surface of residual carbon presented uneven high and low fluctuation, and the whole sample was an irregular network structure. The structure was rough and loose and had some holes with a large pore diameter; however, most of them existed in embedded nonpenetrating form, which is mainly caused by noncombustible gases such as N_2_ and CO_2_ released by polymers after combustion on the carbon layer.

## 4. Conclusions

In this study, the pyrolysis and combustion characteristics of epoxy asphalt were examined using TG-MS, cone calorimeter, and SEM. The main conclusions are as follows:(1)Epoxy asphalt exhibited different characteristics in pyrolysis and combustion stages. When it was close to the peak temperature of each stage, a peak value was observed in the ion flow curve, and the release of volatiles was the largest.(2)The peak HRR, THR, SPR, and TSR of epoxy resin were high, reaching 1015.7 kW/m^2^, 104.5 MJ/m^2^, 0.203m^2^/s, and 2995.2 m^2^/m^2^, respectively, and the combustion was intense. However, the smoke produced from epoxy resin reduced to a certain extent after the base asphalt was added in the epoxy resin in the form of an interpenetrating network. The SPR of epoxy asphalt was high in the early stage, up to 0.220 m^2^/s, then sharply decreased. Moreover, the TSR was close to that of base asphalt.(3)The residual carbon rate was low, and the volatile content was relatively small after the combustion of epoxy resin. Its combustion form is to deepen the combustion layer-by-layer at the pore-forming place. Therefore, the residual carbon surface was dense and flat with few through holes. However, the carbon residue surface of epoxy asphalt after combustion exhibited an irregular network structure, which was rough and loose, with some holes; nevertheless, most of them existed in the form of embedded nonpenetration, which is mainly caused by the impact of noncombustible gases such as N_2_ and CO_2_ released by polymers after combustion on the carbon layer.(4)The results of heat release and smoke release tests showed that epoxy asphalt is not a simple fusion of base asphalt and epoxy resin. Instead, they promote, interact with, and affect each other, and the influence of epoxy resin on the pyrolysis and combustion characteristics of epoxy asphalt was greater than that of base asphalt.

## Figures and Tables

**Figure 1 materials-15-04973-f001:**
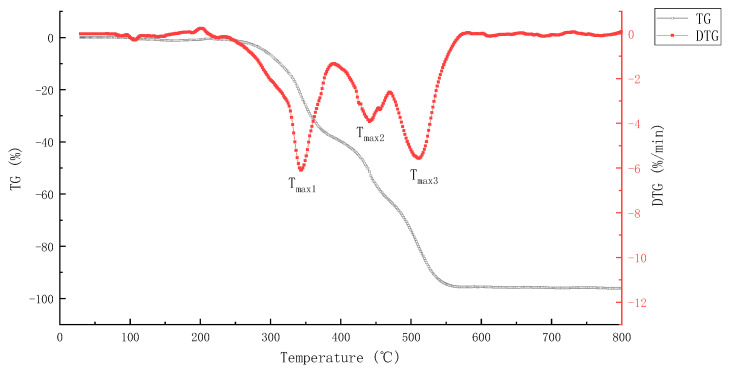
TG and DTG rate curve of epoxy asphalt.

**Figure 2 materials-15-04973-f002:**
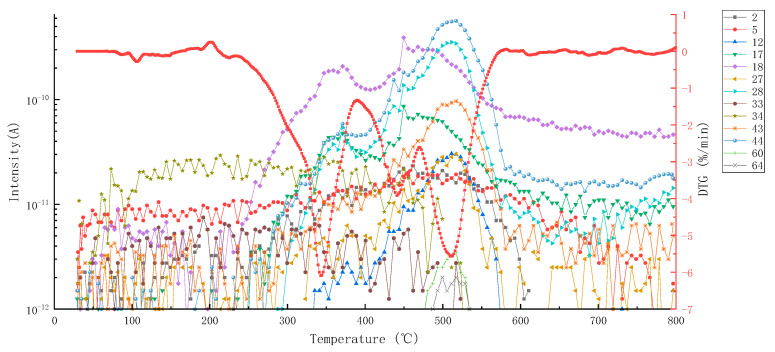
MS-DTG curve of main volatiles from pyrolysis combustion of epoxy asphalt.

**Figure 3 materials-15-04973-f003:**
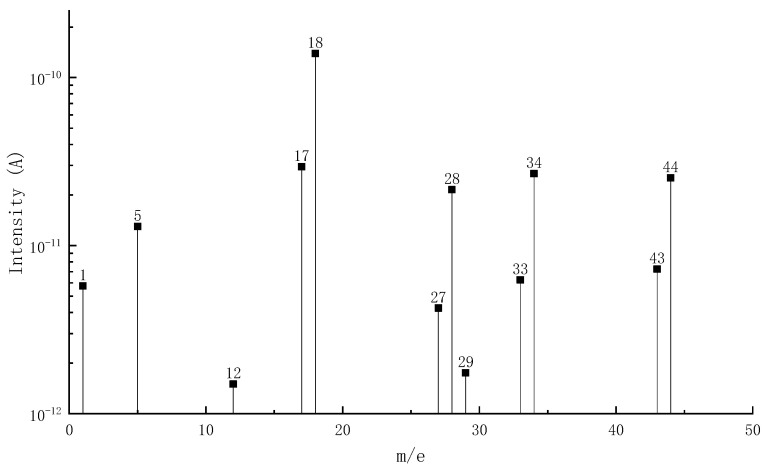
Mass spectrum of volatile matter from pyrolysis combustion of epoxy asphalt at characteristic temperature (342.8 °C).

**Figure 4 materials-15-04973-f004:**
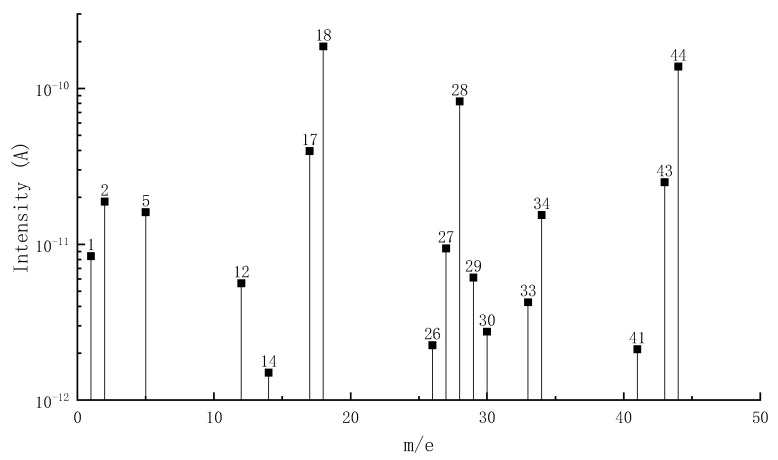
Mass spectrum of volatile matter from pyrolysis combustion of epoxy asphalt at characteristic temperature (440.8 °C).

**Figure 5 materials-15-04973-f005:**
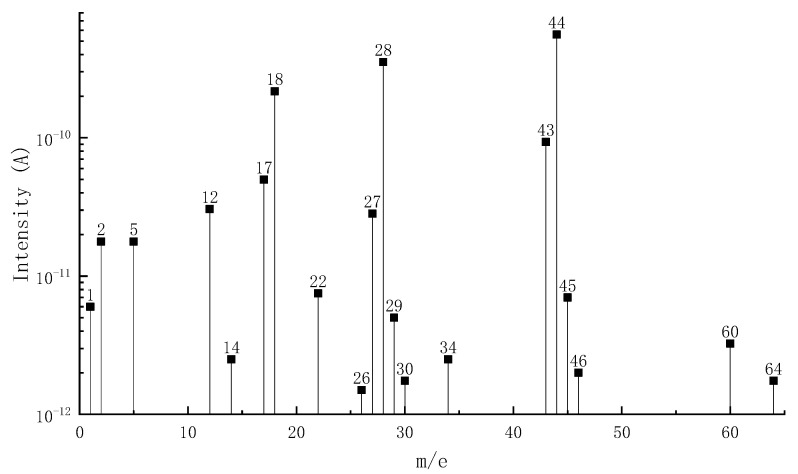
Mass spectrum of volatile matter from pyrolysis combustion of epoxy asphalt at characteristic temperature (510.9 °C).

**Figure 6 materials-15-04973-f006:**
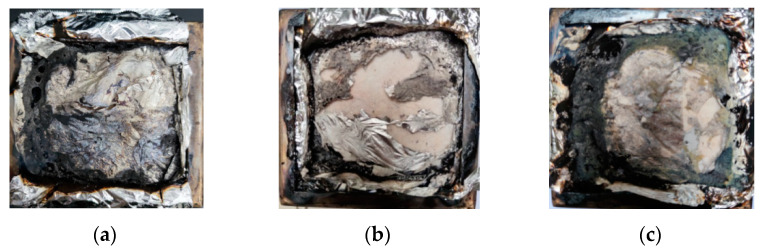
Images of residue after the cone calorimeter test using different materials: (**a**) Base asphalt, (**b**) epoxy resin, and(**c**) epoxy asphalt.

**Figure 7 materials-15-04973-f007:**
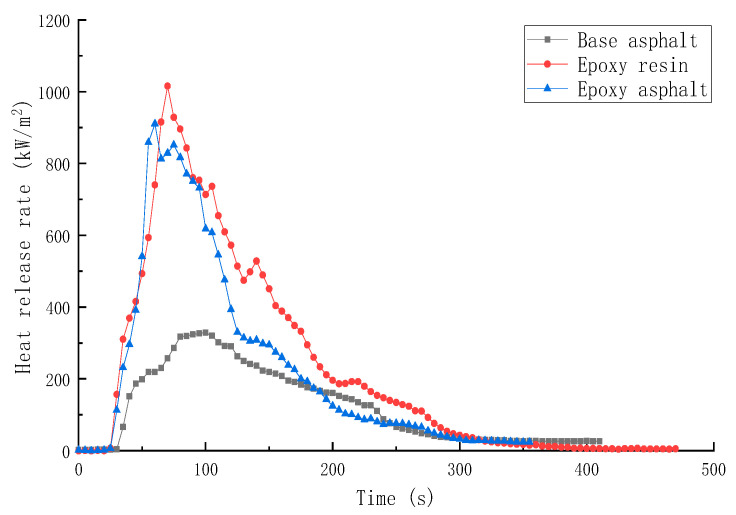
HRR of different materials.

**Figure 8 materials-15-04973-f008:**
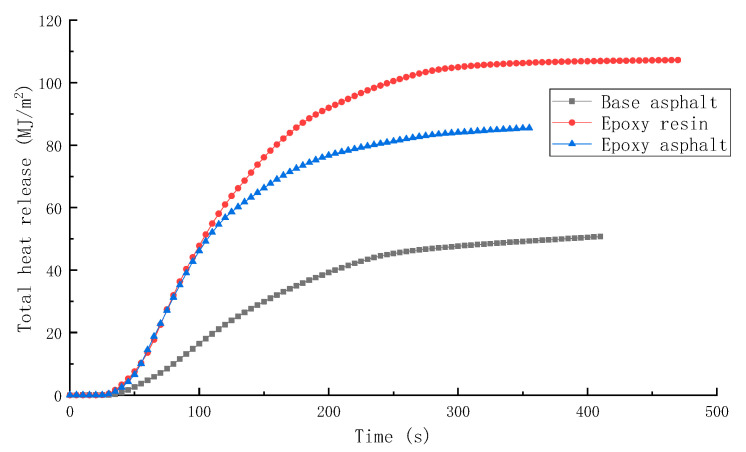
THR of different materials.

**Figure 9 materials-15-04973-f009:**
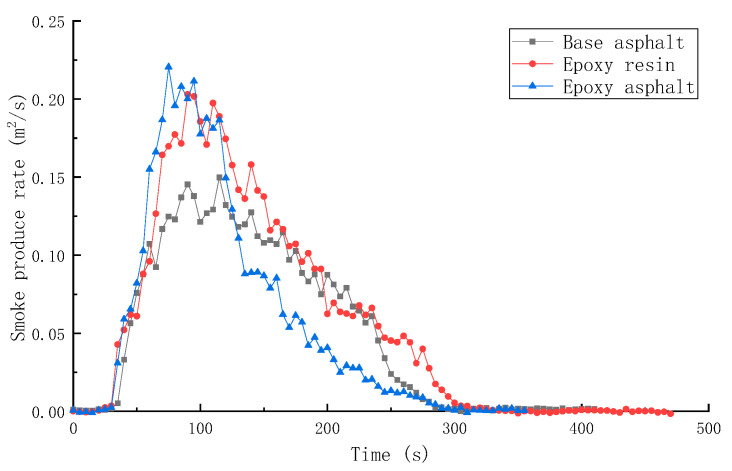
SPR of different materials.

**Figure 10 materials-15-04973-f010:**
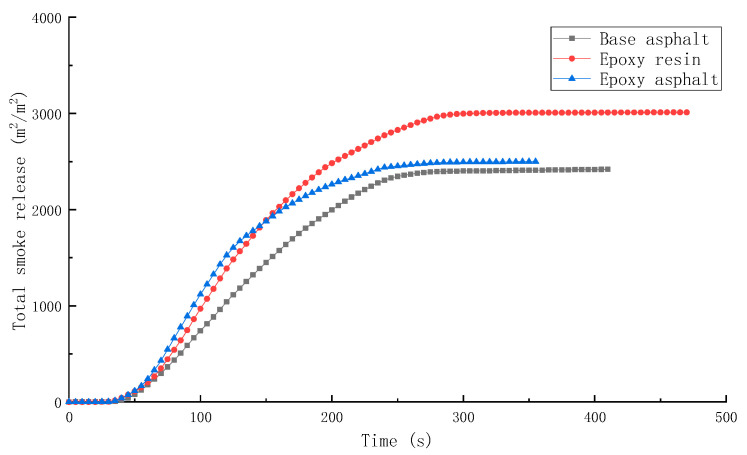
TSR of different materials.

**Figure 11 materials-15-04973-f011:**
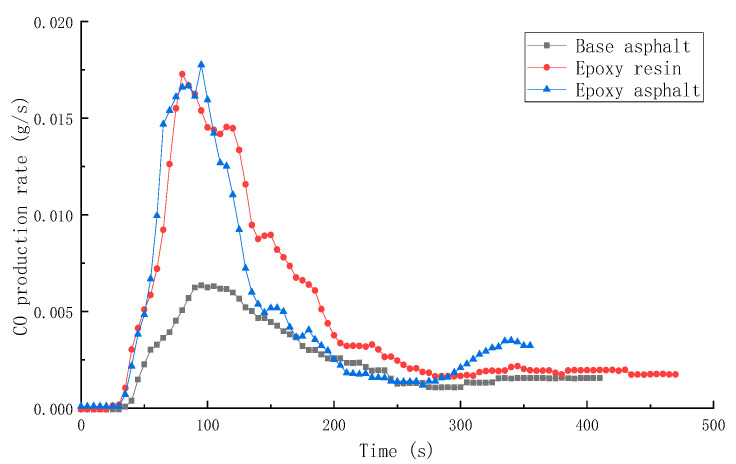
CO production rate (COP) of different materials.

**Figure 12 materials-15-04973-f012:**
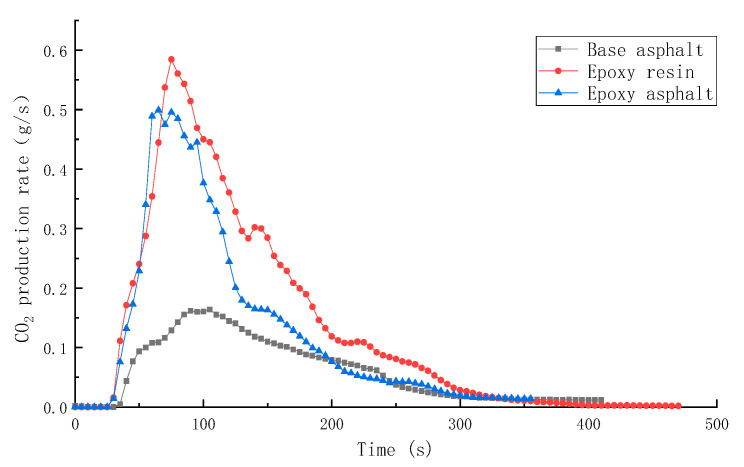
CO_2_ production rate (CO_2_P) of different materials.

**Figure 13 materials-15-04973-f013:**
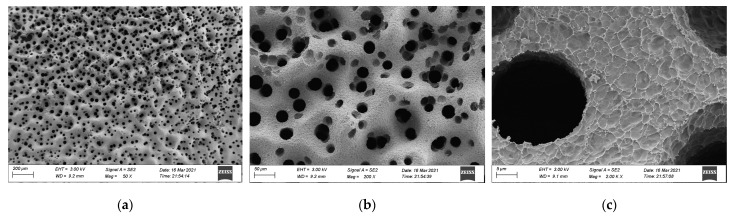
Morphology of residual carbon after the cone calorimeter test of base asphalt: (**a**) 50×, (**b**) 200×, and (**c**) 2000× magnification.

**Figure 14 materials-15-04973-f014:**
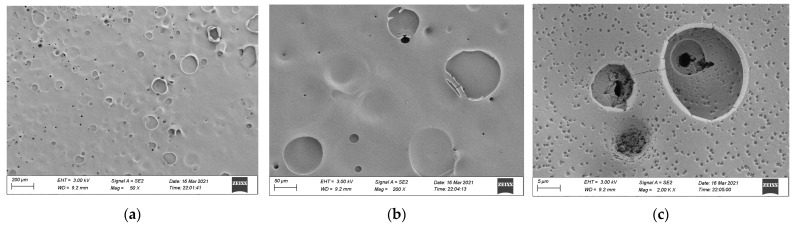
Morphology of residual carbon after the cone calorimeter test of epoxy resin: (**a**) 50×, (**b**) 200×, and (**c**) 2000× magnification.

**Figure 15 materials-15-04973-f015:**
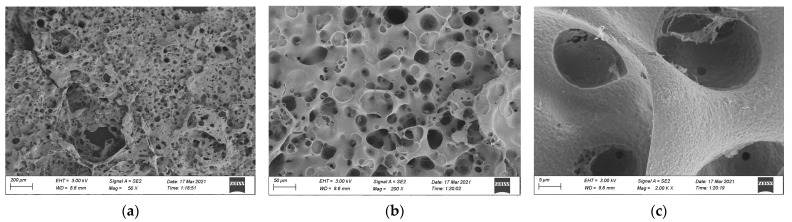
Morphology of carbon residue after the cone calorimeter test of epoxy asphalt: (**a**) 50×, (**b**) 200×, and (**c**) 2000× magnification.

**Table 1 materials-15-04973-t001:** Performance indexes of base asphalt.

Test Items	Pen. /0.1 mm	R&B/°C	Ductility (10 °C) /cm	RTFOT		
				Quality Change/%	Residual Penetration Ratio/%	Residual Ductility (10 °C)/cm
Test value	72	47.5	33	0.48	68.5	9
Technical requirement	60–80	≥45	≥15	≤±0.8	≥61	≥6

**Table 2 materials-15-04973-t002:** Performance indexes of epoxy resin component A.

Test Items	Viscosity (23 °C) /(Pa·s)	Epoxy Equivalent /(g·mol^−1^)	Water Content /%	Flash Point/°C	Density /(g·cm^−3^)	Appearance
Test value	13.278	189	0.02	281	1.164	Transparent
Technical requirement	11–15	185–192	≤0.05	≥200	1.16–1.17	Transparent

**Table 3 materials-15-04973-t003:** Performance indexes of epoxy asphalt.

Test Items	Tensile Strength (23 °C)/MPa	Fracture Elongation (23 °C)/%	Thermoset (300 °C)	Residence Time /min
Test value	4.3	221	Un-melted	98
Technical requirement	≥1.5	≥200	300°C Un-melted	≥40

**Table 4 materials-15-04973-t004:** Distribution of volatile matter from pyrolysis combustion of epoxy asphalt.

Pyrolysis Combustion Stage	Temperature Range/°C	Thermal Weightlessness/%	Main Volatiles (Characteristic Products)
Phase I	232.1–387.9	38.1	H_2_(2), C ion fragment(12), NH_3_(17), H_2_O(18), CNH(27), N_2_(28), CO(28), C_2_H_4_(28), HS-ion fragment(33), H_2_S(34), H_2_O_2_(34), N_3_H(43), CH_2_O_2_(44), C_3_H_8_(44), CO_2_(44), N_2_O(44)
Phase II	387.9–467.3	23.8	H_2_(2), C ion fragment(12), NH_3_(17), H_2_O(18), CNH(27), N_2_(28), CO(28), C_2_H_4_(28), HS-ion fragment(33), H_2_S(34), H_2_O_2_(34), N_3_H(43), CO_2_(44), C_2_H_4_O(44), C_3_H_8_(44), N_2_O(44)
Phase III	467.3–579.9	33.7	H_2_(2), C ion fragment(12), NH_3_(17), H_2_O(18), CNH(27), N_2_(28), CO(28), C_2_H_4_(28), N_3_H(43), CO_2_(44), C_2_H_4_O(44), N_2_O(44), C_3_H_8_(44), C_3_H_8_O(60), C_2_H_4_O_2_(60), C_2_H_8_N_2_(60), C_2_H_4_S(60), SO_2_(64)

**Table 5 materials-15-04973-t005:** Test results of the cone calorimeter test.

Sample	TTI (s)	HRR (kW/m^2^)	THR (MJ/m^2^)	SPR (m^2^/s)	TSR (m^2^/m^2^)	COP (g/s)	CO_2_P (g/s)
Base asphalt	36	328.8	50.5	0.145	2413.9	0.00635	0.1639
Epoxy resin	30	1015.7	104.5	0.203	2995.2	0.01728	0.5845
Epoxy asphalt	31	910.5	85.2	0.220	2492.8	0.01776	0.4991

## Data Availability

Not applicable.

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
