# Peer review of "Pyrolysis Combustion Characteristics of Epoxy Asphalt Based on TG-MS and Cone Calorimeter Test"

_materials, 2022, doi:10.3390/ma15144973_

Round 1

Reviewer 1 Report

Overall, this is a well prepared paper. I have a minor suggestion for the improvement of the paper presentation.

Discussion in sections 3.1.2, 3.2.1, 3.2.2, 3.3 should be written in paragraph, avoid the numbering bullet.

Author should update the paper citation, the latest reference presented was 2020, Should check for the recently published work in 2021 and 2022 

Author Response

Point 1: Discussion in sections 3.1.2, 3.2.1, 3.2.2, 3.3 should be written in paragraph, avoid the numbering bullet.

Response 1: Thanks for your careful review and comments. It have been revised and expressed in the form of paragraphs.

Point 2: Author should update the paper citation, the latest reference presented was 2020, Should check for the recently published work in 2021 and 2022.

Response 2: Thanks for your careful review and comments. The research in this field is relatively few, and the references have been updated by searching the relevant research content in the past two years.

Reviewer 2 Report

In this paper, the heat and smoke release characteristics of epoxy asphalt are shown. These tests are necessary in the conditions of using durable, durable and flammable materials. New extended tests compared to base asphalt are shown. 

Please try to cite more publications from countries other than China.

In the summary, describe in the form of a percentage by number the changes in the examined parameters. This applies to sentences, e.g .:

"The peak HRR, THR, SPR, and TSR of epoxy resin were high, and the combustion was intense."

"However, the smoke produce of epoxy resin reduced to a certain extent after the base asphalt was added in the epoxy resin in the form of interpenetrating network."

"The SPR of epoxy asphalt was high in the early stage, then decreased sharply."

Author Response

Point 1: Please try to cite more publications from countries other than China.

Response 1: Thanks for your careful review and comments. There is relatively little research in the field of pyrolysis and combustion of asphalt and epoxy asphalt, and at present, it is mainly Chinese scholars who are doing research, but the journals are mainly from the United States, Britain and other countries. By searching again, the literature has been supplemented and improved.

Point 2: In the summary, describe in the form of a percentage by number the changes in the examined parameters. This applies to sentences, e.g .:

"The peak HRR, THR, SPR, and TSR of epoxy resin were high, the combustion was intense."

"However, the smoke produce of epoxy resin reduced to a certain extent after the base asphalt was added in the epoxy resin in the form of interpenetrating network."

"The SPR of epoxy asphalt was high in the early stage, then decreased sharply."

Response 2: Thanks for your careful review and comments. The conclusion has been revised as follows:

"The peak HRR, THR, SPR, and TSR of epoxy resin were high, reach 1015.7 kW/m2, 104.5 MJ/m2, 0.203 m2/s, 2995.2 m2/m2, respectively, the combustion was intense. However, the smoke produce of epoxy resin reduced to a certain extent after the base asphalt was added in the epoxy resin in the form of interpenetrating network. The SPR of epoxy asphalt was high in the early stage, up to 0.220 m2/s, then decreased sharply."

Reviewer 3 Report

Attractive manuscript related to the assessment of pyrolysis combustion of an Epoxy Asphalt.

In general, the paper is well presented, but some details must be clarified or corrected/completed:

1.      Lines 74/77: please justify these mixing/test conditions;

2.      Tables 1, 2 & 3: please include the standards used in each test. Where can we find these "Technical requirements”?

3.      Section “2.2. Test Methods (1) TG-MS”: please include the norms used (if you followed some standard);

4.      Figure 1: please include the direction of exothermic peaks;

5.      Lines 141/142: “Figure 3,4, 5 shows …” or “Figures 3,4,5 show …”?

6.      Figure 2: try to simplify this graph;

7.      Table 5 and line 208: please replace “(KW/m2)” with “(kW/m2)”;

8.      Figure 7: please replace “(kw/m2)” with “(kW/m2)”;

9.      Figures 13, 14 & 15: I would replace “(a) 200 μm  (b) 50 μm  (c) 5 μm” with “(a) 50 magnification  (b) 200 magnification  (c) 2000 magnification”;

10.   Lines 263 & 268: please replace “5 μm” with “2000 magnification”;

11.   Where is “discussion” with other results (from other researchers)?

12.   In “References”, you can include the digital object identifier (DOI) for all references where available (as “encouraged” in the “Instructions for Authors”).

13.   Please include references from other sources (those shown are mostly from research carried out in Asia).

Author Response

Response to Reviewer 3 Comments

Point 1: Lines 74/77: please justify these mixing/test conditions.

Response 1: Thanks for your careful review and comments. Component A of epoxy resin is a single substance, not a mixture. In order to avoid ambiguity, the title of Table 2 has been modified. While epoxy asphalt is prepared premixing by component A (E-51 bisphenol A type) and component B (modified aromatic amine curing agent) in a ratio of 60:40 (mass ratio) at 60℃ to first form the epoxy resin, and was then poured into equal mass base asphalt at 150℃ for 4 min shear mixing. The test methods, conditions and technical requirements for performance indexes of epoxy resin and epoxy asphalt shall be implemented in accordance with the specification[22]. It has been modified as required.

[22] General Administration of Quality Supervision, Inspection and Quarantine of the People's Republic of China. General specifications of epoxy asphalt materials for paving roads and bridges. GB/T 30598–2014. Beijing,China.

Point 2: Tables 1, 2 & 3: please include the standards used in each test. Where can we find these "Technical requirements”?

Response 2: Thanks for your careful review and comments. It has been modified as required. Table 1 meet the technical specifications for construction of highway asphalt pavements[22] requirements. Tables 2 & 3 meet the general specifications of epoxy asphalt materials for paving roads and bridges[23] requirements.

[22] Ministry of Transport of the People’s Republic of China. Technical specifications for construction of highway asphalt pavements. JTG F40–2014. Beijing, China.

[23] General Administration of Quality Supervision, Inspection and Quarantine of the People's Republic of China. General specifications of epoxy asphalt materials for paving roads and bridges. GB/T 30598–2014. Beijing,China.

Point 3: Section “2.2. Test Methods (1) TG-MS”: please include the norms used (if you followed some standard).

Response 3:  Thanks for your careful review and comments. The parameters of TG-MS test conditions and methods are determined according to the test equipment and material properties, and there is no fixed specification mode.

Point 4: Figure 1: please include the direction of exothermic peaks.

Response 4: Thanks for your comments. Figure 1 shows the TG and DTG rate curve of epoxy asphalt. Thermogravimetric analysis (TG), which refers to a thermal analysis technology that measures the relationship between the quality of the sample to be tested and the temperature change under the programmed temperature, and is used to study the thermal stability and components of materials. DTG (differential thermal gravity) curve represents the functional relationship between the rate of change of mass with time (dm/dt) and temperature (or time), also known as derivative thermogravimetric analysis. It is a technique to record the first-order derivative of TG curve to temperature or time, taking the rate of change of mass as the ordinate and representing reduction from top to bottom. The abscissa is temperature or time, and from left to right indicates increase. The peaks have been marked in the figure.

Point 5: Lines 141/142: “Figure 3,4, 5 shows …” or “Figures 3,4,5 show …”?

Response 5: Thanks for your careful review and comments. This is a mistaken and has been modified.

Point 6: Figure 2: try to simplify this graph.

Response 6: Thanks for your careful review and comments. Figure 2 shows the MS-DTG curve of pyrolysis volatiles of epoxy asphalt. The image in Figure 2 looks messy. The main reason is that the pyrolysis and combustion of epoxy asphalt is relatively complex, the volatiles changes greatly in different stages, and the TG-MS equipment is sensitive to detect the volatile ion flow, so the figure is irregular. The volatiles with insignificant change has been deleted in this figure, and only the main volatiles is retained. If it continues to be deleted, it will not correspond to the content expressed in Table 4.

Point 7: Table 5 and line 208: please replace “(KW/m2)” with “(kW/m2)”.

Response 7: Thanks for your careful review and comments. This is a mistaken and has been modified.

Point 8: Figure 7: please replace “(kw/m2)” with “(kW/m2)”.

Response 8: Thanks for your careful review and comments. This is a mistaken and has been modified.

Point 9: Figures 13, 14 & 15: I would replace “(a) 200 μm  (b) 50 μm  (c) 5 μm” with “(a) 50× magnification  (b) 200× magnification  (c) 2000× magnification”.

Response 9: Agree. This is more reasonable, and it has been modified as suggestion.

Point 10: Lines 263 & 268: please replace “5 μm” with “2000× magnification”.

Response 10: Agree. This is more reasonable, and it has been modified as suggestion.

Point 11: Where is “discussion” with other results (from other researchers)?

Response 11: Thanks for your careful review and comments. Because there are few relevant studies on epoxy asphalt, there is no comparison and discussion with the studies of other researchers here. Only the test results of this study are analyzed. In order to avoid ambiguity, it has been modified to " Results and Analysis ".

Point 12: In “References”, you can include the digital object identifier (DOI) for all references where available (as “encouraged” in the “Instructions for Authors”).

Response 12: Thanks for your careful review and comments. It has been modified as required.

Point 13: Please include references from other sources (those shown are mostly from research carried out in Asia).

Response 13: Thanks for your careful review and comments. There is relatively little research in the field of pyrolysis and combustion of asphalt and epoxy asphalt, and at present, it is mainly Chinese scholars who are doing research, but the journals are mainly from the United States, Britain and other countries. By searching again, the literature has been supplemented and improved.
